# Mathematical Computations of Long-Term Settlement and Bearing Capacity of Soil Bases and Foundations near Vertical Excavation Pits

**Zaven G. Ter-Martirosyan, Armen Z. Ter-Martirosyan \*** and **Yulia V. Vanina**

Department of Soil Mechanics and Geotechnical Engineering, National Research Moscow State Civil Engineering University, 26, Yaroslavskoye Shosse, 129337 Moscow, Russia
* Correspondence: gic-mgsu@mail.ru

**Abstract:** The present paper describes and provides an analytical solution for the problem of evaluating the settlement and load-bearing capacity of weighty soil layers of limited thickness resting upon incompressible soil bases and an excavation pit wall, upon exposure of the foundation to a distributed load in the vicinity of a wall. The authors developed a method for determining the stressed state component in the reduced engineering problem based on the Ribere–Faylon trigonometric series, accounting for the nonlinear deformation properties of soils. To determine the settlement over time of the foundation near the pit, we used the A.Z. Ter-Martirosyan's model to describe shear deformations and the Kelvin–Voigt model to describe volume deformations, assuming that $\varepsilon \cdot_z(t) = \varepsilon \cdot_v(t) + \varepsilon \cdot_\gamma(t)$, according to the Hencky's system of physical equations. The obtained solutions make it possible to assess the long-term deformation of soil bases and the long-term load-bearing capacity with respect to nonlinear rheological properties in a way that accurately corresponds to the actual performance of subsoils exposed to loading. The theoretical results were followed by numerical experiments to prove their validity.

**Keywords:** stress–strain state; Ribere-Faylon trigonometric series; A.Z. Ter-Martirosyan's rheological model; Kelvin-Voigt rheological model; long-term stability of the soil base; creep curves

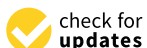



## 1. Introduction

It is known that in applied soil mechanics, one of the critical problems of high-rise constructions with a substantial subsurface part is the quantitative assessment of the stress–strain statement of a soil basis that has a deep excavation pit with an enclosure and substructures, subject to engineering and geological conditions. When the substructure of a high-rise building interacts with the surrounding soil basis outside the excavation pit enclosure and the layer below the foundation, there arises complex heterogeneous stress–strain state, which changes in space and time throughout the construction and operation of the building. Particular difficulties arise when the soil basis is heterogeneous and demonstrates rheological properties and if there is an additional source of loading near the excavation pit enclosure.

In the present paper, the authors propose an analytical solution for a problem of stress–strain state of a weighty soil layer of a limited thickness (*h*) resting upon an incompressible soil basis and an excavation pit wall on exposure to the foundation, having width *b = 2a*, with the distributed load *q = const* (kPa) in the vicinity of the wall (see Figure 1). To evaluate the settlement over time and the long-term stability of the soil basis, the authors applied the elastic–viscous model, developed by A.Z. Ter-Martirosyan [1] to describe shear deformations, and the Kelvin—Voigt model [2] to describe volumetric deformations. According to H. Hencky [3], any deformation can be represented as the sum of its volumetric and shear components, i.e., $\varepsilon_z = \varepsilon_v + \varepsilon_\gamma$, where $\varepsilon_v = f(\sigma_m)$, and $\varepsilon_\gamma = f(\tau_i, \sigma_m)$.

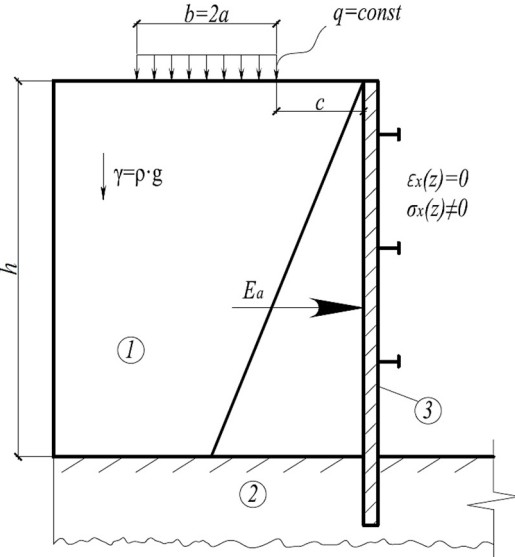

**Figure 1.** Computational model of interaction between a weighty layer (1), having thickness (*h*) and resting on an incompressible soil base (2), with a vertical fixed wall of an excavation pit enclosure (3) subjected to a distributed loading *q* = *const* along strip *b* = 2*a* at a distance (*c*) from the enclosure.

Attempts to quantify the stress–strain state of a weighty soil basis in the quarter-plane were made by O.Y. Shekhter [4], V.A. Florin [5], M.I. Gorbunov-Posadov [4], Z.G. Ter-Martirosyan [6] and others. It is noteworthy that Z.G. Ter-Martirosyan obtained the analytical solution to the problem of stresses in areas with curvilinear boundaries, including deep pits. He used the Kolosov and Muskhelishvili method of complex potentials by mapping the upper area with a curvilinear boundary onto the lower half-plane.

Z.G. Ter-Martirosyan [7,8] successfully solved the problem of identifying the stress–strain state of the half-plane, if the boundary of its limited thickness layer was subjected to loading; this problem was successfully solved using the Ribere–Faylon trigonometric series. The solution allows using MathCAD software package to solve the problem of stress–strain state both for a layer of limited width and for a layer resting on an incompressible base. Methods to determine the stress–strain state of a soil basis subjected to loads have been developed [9–11]. Problems and techniques for excavation pits were also described [12–16].

In 1925, K. Terzaghi made the first attempt to take account of the rheological properties of soils to determine creep deformations [17]. Some researchers, including N.N. Maslov [18], S.S. Vyalov [2], V.A. Florin [5], M.N. Gol'dshtejn [19], Y.K. Zareckij [20], S.R. Meschyan [21,22], G.I. Ter-Stepanyan [23], N.A. Cytovich [24], Z.G. Ter-Martirosyan [24–26], A.Z. Ter-Martirosyan [1,26,27], as well as foreign scientists, including L. Shukle [28] and others [29–34] contributed greatly to the study of deformations of clay soils and their influence on projected changes in the stress–strain state of soil bases. Most of these researchers focused on developing the theory of creep and making new rheological models. Y.K. Zareckij, A.L. Goldin, Z.G. Ter-Martirosyan and A.Z. Ter-Martirosyan made a great contribution to the theories of consolidation and creep. They solved one-dimensional, two-dimensional and three-dimensional problems of consolidation, taking into account the creep of the soil skeleton.

In this work, the authors offer an improved model, originally developed by A.Z. Ter-Martirosyan [1], to describe shear deformations, and another improved model, developed by Kelvin and Voigt [2], to describe volumetric deformations, assuming that $\varepsilon \cdot_z(t) = \varepsilon \cdot_v(t) + \varepsilon \cdot_\gamma(t)$ according to the Hencky's system of physical equations [3]. Earlier, many researchers, including S. S. Vyalov [2], A. R. Rzhanicyn [35], had assumed that the creep curve γ-t had a double curvature, including initial (decaying), intermediate (steady), and final (evolving) stages, with a possible transition to collapse. However, they did not manage to describe this curve by a single equation. Hence, A.Z. Ter-Martirosyan was the world's first developer of a new rheological model (2016) to describe the γ-t creep curve

with a double curvature. The author's rheological model was also verified by laboratory tests carried out by A. Z. Ter-Martirosyan, L. Yu. Ermoshina and A. Manukyan. [27] Further, this model was applied by Z.G. Ter-Martirosyan and A.Z. Ter-Martirosyan to solve the boundary problem with a distributed loading [1]. Mathematical formulas for the settlement and load-bearing capacity of a soil base adjacent to an excavation pit were also developed, taking into account the elastoplastic properties of the soil [26].

The Boltzmann–Voltaire theory of hereditary creep for elastic media, proved by N.Kh. Arutyunyan [36], as well as other studies, to be subjected to the constancy of the Poisson's coefficient, was used to determine the settlement over the time and the long-term bearing capacity of the soil base. In a first approximation, it was assumed that $\nu_e \approx \nu_v$.

## 2. Materials and Methods

The action of a distributed load $q$ = const on a horizontal section of width $b = 2a$ at a distance (c) from the edge of an enveloped structure with a rectangular profile on a soil basis resting upon an incompressible soil layer was determined. It was assumed that a vertical wall was fixed by struts, whereas vertical shifts of the soil were permitted (see Figure 1). We assumed the following boundary conditions on the upper and lower boundaries of the array area:

$$at\ y = 0\ and\ y = 2h:\ \sigma_y(x,0) = \sigma_y(x,2h) = q$$
$$(-a \le x \le a);\ \sigma_y(x,0) = \sigma_y(x,2h) = 0(-l \le x \le -a)\ and\ (a \le x \le l); \quad (1)$$
$$at\ y = 0\ and\ y = 2h:\ \tau_{xy}(x,0) = 0;\ \tau_{xy}(x,2h) = 0$$

Horizontal displacements on the right boundary $x = \pm l$ were absent. We considered another boundary condition in the form:

$$u(\pm l) = 0\ \nu(\pm l) \ne 0 \quad (2)$$

Components of the stress state of the soil basis outside the enclosing structure were calculated using the method of Z.G. Ter-Martirosyan [7,8] applied to the Ribere-Faylon trigonometric series in the following manner:

$$\sigma_z(x,z) = \frac{qb}{l} +$$
$$+ \frac{4q}{\pi} \sum_{m=1}^{\infty} \frac{\sin\frac{m\pi b}{l}}{m} \left[ \frac{\left(\frac{m\pi h}{l}ch\frac{m\pi h}{l} + sh\frac{m\pi h}{l}\right)ch\frac{m\pi(z-h)}{l} - \frac{m\pi(z-h)}{l}sh\frac{m\pi(z-h)}{l}sh\frac{m\pi h}{l}}{sh\frac{2m\pi h}{l} + \frac{2m\pi h}{l}} \right] \cos\frac{m\pi x}{l} \quad (3)$$

$$\sigma_m(x,z) = \frac{1+\nu}{3} \left[ \frac{qb}{l}\frac{1}{1-\nu} + \frac{8q}{\pi} \sum_{m=1}^{\infty} \frac{\sin\frac{m\pi a}{l}}{m} \frac{sh\frac{m\pi b}{l}ch\frac{m\pi(z-h)}{l}}{sh\frac{2m\pi h}{l} + \frac{2m\pi h}{l}} \cos\frac{m\pi x}{l} \right] \quad (4)$$

where:

$q$—distributed load, kPa;

$b$—width of the strip of distributed load, m;

$l$—width of the estimated area;

$m$—any integer. m = 1;

$h$—depth of the estimated area;

$\nu$—Poisson 's ratio

According to the results of clay soils tests, conducted by S.S. Vyalov, S.R. Meschyan and others [2,21,22] (see Figure 2), the diagram of non-reducing creep deformations is typical for soils having rheological properties and subjected to high stresses with long-term loading (see Figure 3).

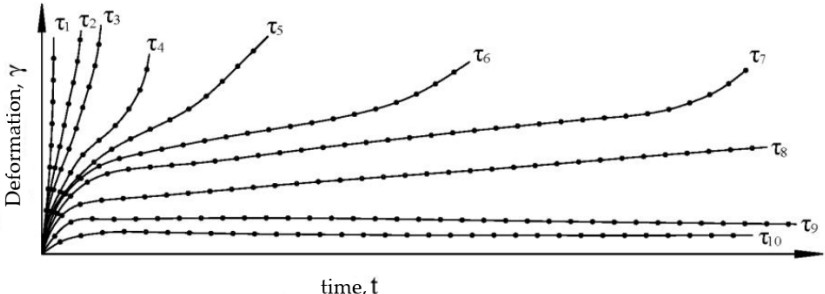

**Figure 2.** Curves describing the shear creep of clay according to S.S. Vyalov, 1978. [2] ($\tau_s$ is the limiting value of tangential stresses (shear stresses)).

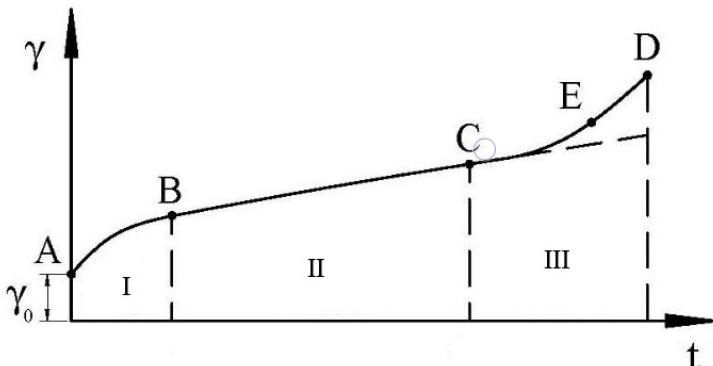

**Figure 3.** Diagram showing how deformations change over time: Stage I is the stage of steady creep (section AB); Stage II is the stage of a steady flow process (section BC), Stage III is the stage of progressive collapse (section CE). [2].

In 2016, A. Z. Ter-Martirosyan [1] proposed a new rheological equation allowing to plot the $\gamma$–$t$ relationship at different $\tau$ as a double-curvature curve (Figure 4); the equation encompassed several stages: initial, nonlinear, intermediate with a steady shear rate, and final, characterized by a growing rate and a tendency to progressive collapse.

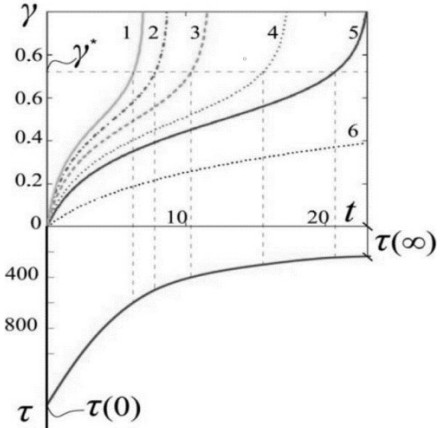

**Figure 4.** Creep curves $\gamma$–$t$ at $\tau1 > \tau2 > \cdots \tau5$ (the top part) and the long-term strength curve (the bottom part) $\tau(0) \rightarrow \tau(\infty)$ at $t \rightarrow \infty$, obtained using Formula (3).

When $\tau = const$, the rheological model developed by A.Z. Ter-Martirosyan [1] can be described by the following equation:

$$\gamma = \frac{\tau - \tau^*}{\eta_\gamma(\sigma_m)} \left( \frac{e^{-\alpha\gamma}}{a} + \frac{e^{\beta\gamma}}{b} \right) \tag{5}$$

where $\tau$ and $\tau^*$ are the effective and the limiting value of tangential stresses applied to the soil specimen; $\gamma \cdot (\gamma)$ is the rate of angular deformation depending on deformation $\gamma$. The expression in parentheses represents the function of simultaneous hardening and softening, where $\gamma$ is the best measure of hardening according to Y.N. Rabotnov [37].

$\eta_\gamma(\sigma_m)$ is the initial shear viscosity of the soil, which generally depends on the mean stress $\sigma_m$.

$\alpha$, $\beta$, $a$ and $b$ are the hardening (softening) parameters of a clay soil, which are determined by the kinematic shear ($\gamma \cdot = const$), presented in Figure 2.

According to this model (Formula (5)), the shear strain rate depends nonlinearly on the accumulated shear strain $\gamma \cdot (\gamma)$. Formula (5) can be applied to construct $\tau$–$t$ curves for the kinematic shear ($\gamma \cdot = const$), as well as relaxation curves $\tau$ (0) $\rightarrow$ $\tau$ (t) at $\gamma$ (0) = const (see Figures 4 and 5).

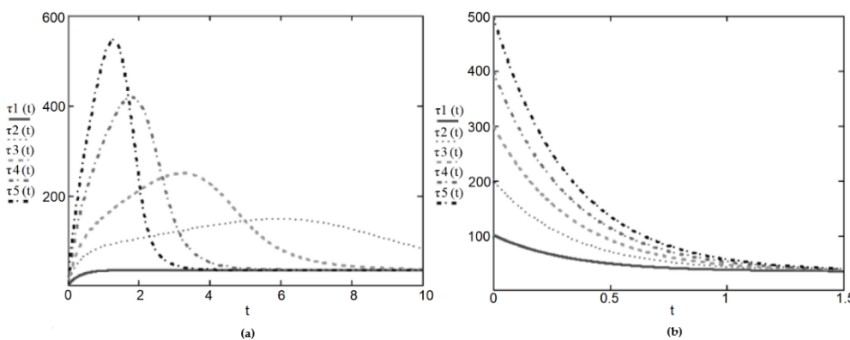

**Figure 5.** (**a**) $\tau$–$t$ curves made using the kinematic testing results at different values of $\gamma = const$, $\gamma_1 > \gamma_2 > \cdots \gamma_5$; (**b**) shear stress relaxation curves $\tau$ (t) at different initial shear stresses $\tau(0)$ and $\gamma = const$.

Hence, one function can be used to obtain all three types of rheological curves, including curves for creep, represented as $\gamma$–$t$ curves at different $\tau$–$\tau^* = const$ (see Figure 4), kinematic shear $\tau$–$t$ at different ($\gamma \cdot = const$) (see Figure 5a), and relaxation $\tau$–$t$ at different initial $\tau$ (0) and $\gamma$ (0) = const (see Figure 5b).

It is noteworthy that in many cases these curves, determined using the new rheological equation (Formula (5)), coincided with the curves obtained as a result of laboratory testing conducted by different researchers [2,18–23,27]. It is also important that these curves were drawn using the same rheological parameters ($\alpha$, $\beta$, $a$ and $b$). Therefore, the authors' approach has a significant advantage over the technology of A.R. Rzhanicyn [35], who recommended to plot the double-creep curvature (see Figure 3) in parts.

The system of H. Hencky's physical equations [3], that allows determining linear and non-linear dependences between stresses and strain rates, has the following form:

$$\dot{\varepsilon}_x = \dot{\chi}(\sigma_x - \sigma_m) + \dot{\chi}^* \cdot \sigma_m; \dot{\gamma}_{xy} = 2\dot{\chi} \cdot \tau_{xy} \tag{6}$$

$$\dot{\varepsilon}_y = \dot{\chi}(\sigma_y - \sigma_m) + \dot{\chi}^* \cdot \sigma_m; \dot{\gamma}_{yz} = 2\dot{\chi} \cdot \tau_{yz} \tag{7}$$

$$\dot{\varepsilon}_z = \dot{\chi}(\sigma_z - \sigma_m) + \dot{\chi}^* \cdot \sigma_m; \dot{\gamma}_{zx} = 2\dot{\chi} \cdot \tau_{zx} \tag{8}$$

where

$$\dot{\chi} = \frac{\dot{\gamma}_i}{2\tau_i} = \frac{f(\tau_i, \sigma_m, \mu_\sigma, t)}{2\tau_i}; \tag{9}$$

$$\dot{\chi}^* = \frac{\dot{\varepsilon}_m}{\sigma_m} = \frac{f^*(\tau_i, \sigma_m, \mu_\sigma, t)}{2\tau_i}; \tag{10}$$

$$\dot{\chi} = \frac{\dot{\gamma}_i}{2\tau_i} = \frac{1}{2\eta_\gamma(\sigma_m)} \cdot \left( \frac{e^{-\alpha\varepsilon_z}}{a} + \frac{e^{\beta\varepsilon_z}}{b} \right); \tag{11}$$

where $\varepsilon_x$, $\varepsilon_y$, $\varepsilon_z$, $\sigma_x$, $\sigma_y$, $\sigma_z$ are components of strain rates and normal stresses for the axes x, y, z, respectively; $\sigma_m$ is the mean stress, $\varepsilon_m$ is the mean deformation;

$\gamma_{xy}$, $\gamma_{yz}$, $\gamma_{zx}$, $\tau_{xy}$, $\tau_{yz}$, $\tau_{zx}$ are the angular velocities of deformation and shear stresses for the directions x–y, y–z, z–y, respectively;

$\chi$, $\chi^*$, $\mu_\sigma$ are parameters of shear and volumetric viscous deformation.

The Kelvin–Voigt viscoelastic model [2] was used as the computational model to determine nonlinear volumetric deformations, taking into account the rheological properties of soil:

$$\sigma_m = \sigma_m^{elast} + \sigma_m^{visc} = (\sigma_m) \cdot \varepsilon_m + \eta_v \cdot \dot{\varepsilon}_m \tag{12}$$

Which, if $\varepsilon_m$ $(t \approx 0) = 0$, leads to the following expression:

$$\varepsilon_m(t) = \frac{\sigma_m}{(\sigma_m)} \cdot \left(1 - e^{\frac{-K \cdot t}{\eta_v}}\right) \tag{13}$$

where $\eta_v$ is the volumetric viscosity, $t$ is time, $K$ is the volumetric deformation modulus.

The volumetric deformation rate will decrease if $t \to 0$, parameter $\varepsilon(t) \to 0$,:

$$\dot{\varepsilon}_m(t) = \frac{\sigma_m}{(\sigma_m)} \cdot \left(\frac{-K}{\eta_v} \cdot e^{\frac{-K}{\eta_v}}\right) \tag{14}$$

Relationships between stresses and strain rates were identified using the Hencky's system of physical equations [3], which describes the strain rate as the sum of volumetric and shear strain rates ($\varepsilon \cdot_z = \varepsilon \cdot_\gamma + \varepsilon \cdot_v$) as follows:

$$\dot{\varepsilon}_z = \frac{\sigma_z - \sigma_m}{\eta_\gamma(\sigma_m)} \cdot \left(\frac{e^{-\alpha \varepsilon_z}}{a} + \frac{e^{\beta \varepsilon_z}}{b}\right) + \frac{\sigma_m}{(\sigma_m)} \cdot \left(\frac{-K}{\eta_v} \cdot e^{\frac{-K}{\eta_v}}\right) \tag{15}$$

## 3. Results

Calculations using the Formulas (3)–(4) performed using the software complex Math-CAD enabled to determine the components of stresses along the entire plane at $z > 0$ and $\pm x$, according to the computational scheme (Figure 2). The stress component for $q = 200$ kPa, $c = 6$ m, $b = 2a = 6$ m are presented in Figure 6a,b. For comparison purposes, a similar calculation was performed for a quarter of the plane with the boundary condition on the $z$ axis $\varepsilon_x = 0$, $\sigma_x \neq 0$ using PLAXIS 2D software package. The given values were randomly selected based on the previous design of similar structures in order to quantify the proposed solution. The results are presented in Figure 6.

A calculation in the PS PLAXIS was carried out to verify the isolines of vertical and mean stresses obtained earlier by analytical calculation using Formulas (1)–(3). We used the model Linear Elastic in PS PLAXIS, with the parameters E = 25,000 kPa, $\nu$ = 0.37, $\gamma$ = 0 kN/m$^3$.

The following parameters (see Table 1) were used to obtain the curves describing the creep of compressed soil base layers over time (Formula (15)):

**Table 1.** Soil base parameters.

| $\eta_\gamma$, kPa Day | K, kPa | K/ηv, 1/Day | a | b | $\alpha$ | $\beta$ |
|---|---|---|---|---|---|---|
| $1.157 \times 10^5$ | 3500 | 0.0043 | 1.2 | 60 | 171 | 40 |

The strain–time graphs, obtained using Formula (15) for the layer located at a depth of $z = 4$ m, with $\sigma_z$ = 183.92 kPa and $\sigma_m$ = 101.57 kPa, are shown in Figure 7.

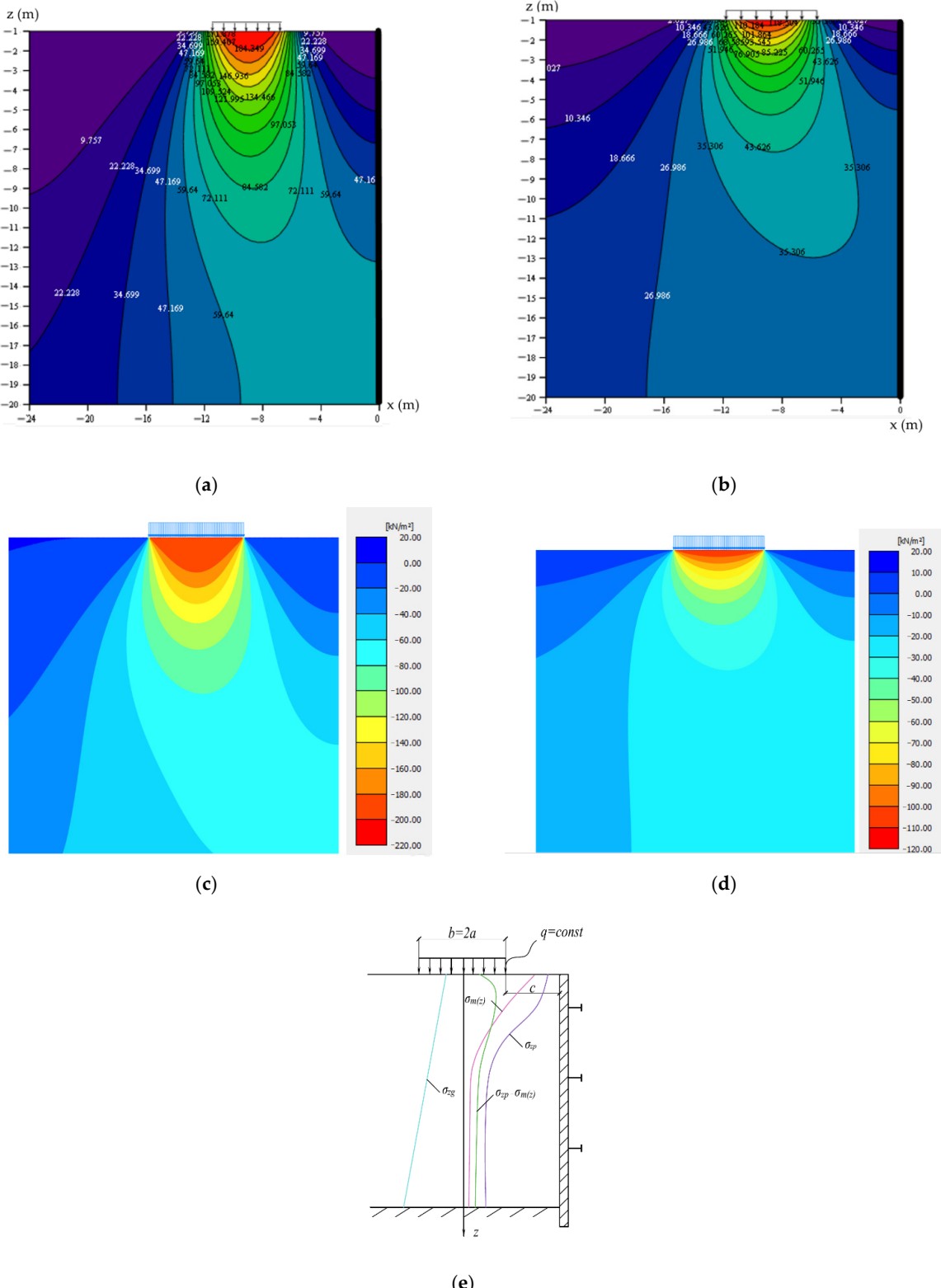

**Figure 6.** Results of analytical and numerical calculations of stress components at $q$ = 200 kPa, $c$ = 6 m, $b = 2a$ = 6 m. Isolines of vertical stresses $\sigma_z$ obtained using MathCAD software package (**a**); isolines of mean stresses $\sigma_m$ obtained using MathCAD software package (**b**); isolines of vertical stresses $\sigma_z$ obtained using PLAXIS 2D software package (**c**); isolines of mean stresses $\sigma_m$ obtained using PLAXIS 2D software package (**d**); the calculation pattern applied to determine shear and volumetric deformations of the soil basis based on the physical equations derived by H. Hencky, including $\sigma_{zp}$, $\sigma_{zg}$, $\sigma_{zm}$ epures (**e**).

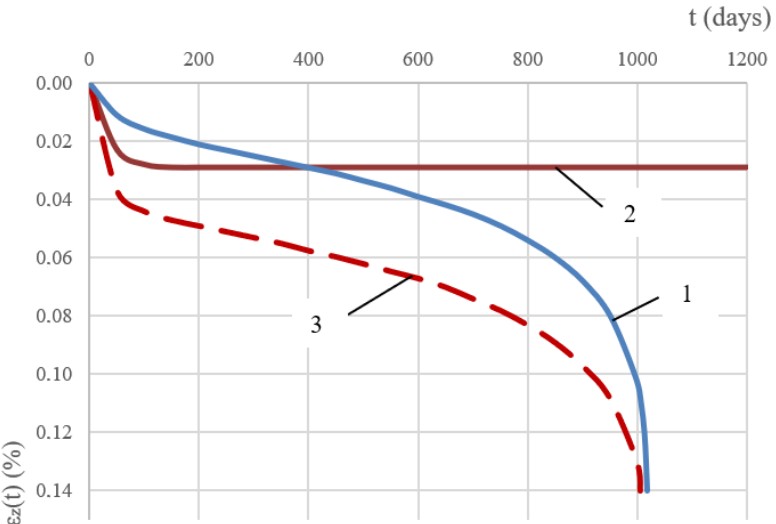

**Figure 7.** Shear deformations in a layer $\varepsilon_z$–$t$: 1: determined using the Ter-Martirosyan model; 2: volumetric deformations determined using the Kevin–Voigt model; 3: total deformations determined using Formula (14).

Similar results were obtained for other layers of the compressible soil base. In the previous paragraphs of this article, formulas for determining the settlement of a layer, having finite thickness ($h = 20$ m), were provided as the sum $\varepsilon_{zi}(t) = \varepsilon_{zi}(\gamma,t) + \varepsilon_{zi}(\varepsilon,t)$. Hence, we could determine the total settlement of the compressible soil base:

$$\sum S'(t) = \sum S_\gamma(t) + \sum S_v(t) \tag{16}$$

where $\sum S_v(t)$ is he sum of volumetric deformations, $\sum S_\gamma(t)$ is the sum of shear deformations.

The calculations, performed to find the total settlement, are shown in Figure 8.

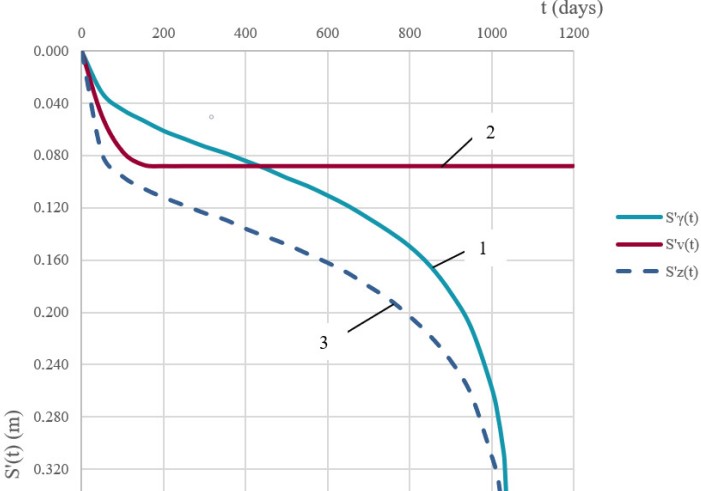

**Figure 8.** Soil base settlements $S'(t)$–$t$: 1: determined using the Ter-Martirosyan model; 2: determined using the Kevin–Voigt model; 3: total deformations determined using Formula (15).

The soil base settlement over time, calculated using the Kelvin–Voigt model [2] and the A.Z. Ter-Martirosyan model [1], included in the Hencky's system of physical equations [3], could be represented with the double-curvature curve settlement–time ($S'$–$t$); at a certain moment in time, it switched to the stage of progressive settlement ($S'(t) \to \infty$; $t = t^*$ is the time of collapse).

## 4. Discussion

We obtained graphs describing the $S'$–$t$ dependence for different values of loading on the foundation, $p = q$ (Figure 9a). The graph showing the long-term stability of the foundation is shown in Figure 9b. The loading values were: $p_1 = 100$ kPa; $p_2 = 150$ kPa; $p_3 = 180$ kPa; $p_4 = 200$ kPa; $p_5 = 220$ kPa; $p_6 = 250$ kPa.

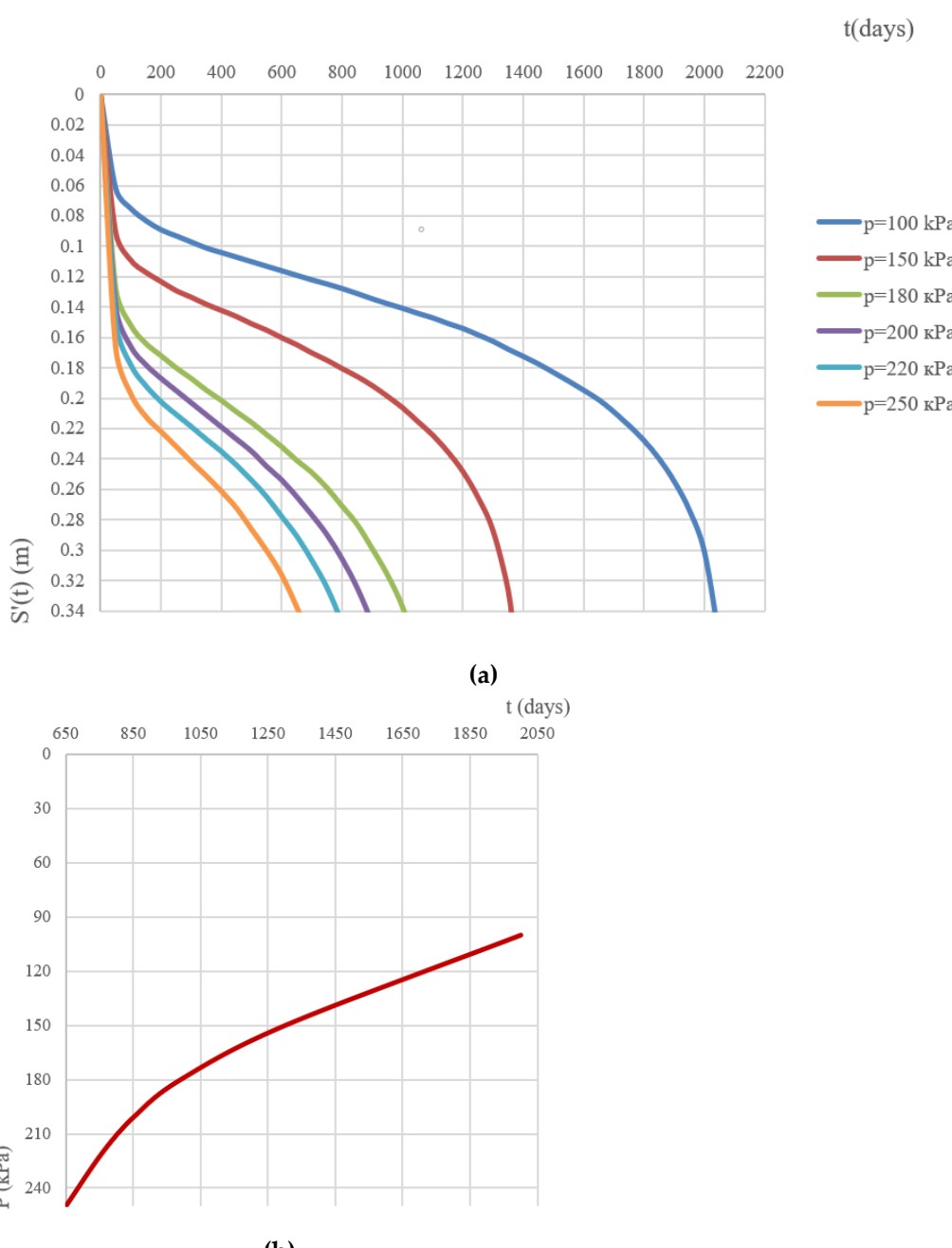

(a)

(b)

**Figure 9.** Total soil base settlements S–t for different loadings from the foundation p (**a**); graph of the long-term stability of the soil base (**b**).

Based on the obtained $S'$(t)–t dependences presented in Figure 9, it appears that the larger the load $p = q = const$, the faster the soil base enters the stage of progressive collapse.

The main advancement of this study is the development of a method for determining the stressed state component in a reduced engineering problem based on the Ribere–Faylon trigonometric series, accounting for the nonlinear deformation properties of soils. The use of the authors' rheological model and the Kelvin–Voigt model allows describing creep

deformations as the sum of shear creep deformations and volumetric creep deformations, assuming that $\varepsilon \cdot_z(t) = \varepsilon \cdot_v(t) + \varepsilon \cdot_\gamma(t)$, according to the Hencky's system of physical equations. A. Z. Ter-Martirosyan proposed a new rheological equation allowing to plot the $\gamma$–$t$ dependence at different values of $\tau$ as a double-curvature curve. The author's rheological model was verified by laboratory tests, which were carried out by A. Z. Ter-Martirosyan, L. Yu. Ermoshina and A. Manukyan [27]. This dependence was verified by clay soils tests, conducted by S.S. Vyalov, S.R. Meschyan and others in the last century.

## 5. Conclusions

Summing up the obtained results, the following can be concluded:

- The selected geomechanical soil base model (its geometric parameters and initial and boundary conditions), as well as the computational model of the soil ground (linear, nonlinear and rheological) and the type of physical equations used (Hooke's system and Hencky's system), have a significant impact on the type of settlement–load curve ($S^{'}(t)$–t) and on the long-term stability of a soil basis.
- The authors' analytical method for the quantitative evaluation of the settlement of soil bases and foundations in time was developed for adjacent built-up areas outside the enclosure of an excavation pit.
- The computational model used in this research along with the rheological model of A.Z. Ter-Martirosyan for shear and the volumetric soil deformation model of Kelvin–Voigt used as part of the physical equations derived by H. Hencky, allowed determining soil deformations $\varepsilon(\sigma$–$\varepsilon, \tau$–$\gamma)$, represented as the sum of volumetric and shear components of the deformations ($\varepsilon_z = \varepsilon_{z,v} + \varepsilon_{z,\gamma}$). In this case, the deformation–time curve ($\varepsilon_z(t)$—$t$) had a double curvature.
- The joint application of the models, developed by A.Z. Ter-Martirosyan and Kelvin–Voigt allows obtaining a graph describing the long-term bearing capacity of a soil base for various loads $p = q = const$.

**Author Contributions:** Conceptualization, methodology, investigation, software, Z.G.T.-M. and A.Z.T.-M.; formal analysis, Y.V.V. and A.Z.T.-M.; writing—original draft preparation, Z.G.T.-M. and A.Z.T.-M.; writing—review and editing, all authors; visualization, Y.V.V.; supervision, Z.G.T.-M. and A.Z.T.-M. All authors have read and agreed to the published version of the manuscript.

**Funding:** This work was financially supported by the Ministry of Science and Higher Education (grant number 075-15-2021-686). All tests were carried out using research equipment at the Head Regional Shared Research Facilities of the Moscow State University of Civil Engineering.

**Institutional Review Board Statement:** Not applicable.

**Informed Consent Statement:** Not applicable.

**Data Availability Statement:** The data used to support the findings of this study are included within the article. The original details of the data presented in this study are available on request from the corresponding author.

**Conflicts of Interest:** The authors declare no conflict of interest.

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
