# Peer review of "Mathematical Computations of Long-Term Settlement and Bearing Capacity of Soil Bases and Foundations near Vertical Excavation Pits"

_axioms, doi:10.3390/axioms11120679_

Round 1
Reviewer 1 Report
This manuscript presents an analytical solution for settlement and bearing capacity of foundations. It is clear that the topic has significances of engineering practice for foundation and pit engineering. The research is based on solid verification, then clarifies the critical parameters, so the conclusions are reliable.
There is one suggestion that the introduction part should add more literature review on the current progress of the related study and emphasize on the innovation and significance of the current work.
Fig.2, Fig.3 are not clear and should be reproduced.
Author Response
This manuscript presents an analytical solution for settlement and bearing capacity of foundations. It is clear that the topic has significances of engineering practice for foundation and pit engineering. The research is based on solid verification, then clarifies the critical parameters, so the conclusions are reliable.
There is one suggestion that the introduction part should add more literature review on the current progress of the related study and emphasize on the innovation and significance of the current work.
Thank you for your review. It is stated that author’s rheological model makes it possible to describe the shear creep deformations as γ-t creep curve with double curvature. There is no other rheological model which is makes it possible as much as this model makes. Also, author’s rheological model was verified by laboratory tests, which was carried out by A. Z. Ter-Martirosyan, L. Yu. Ermoshina and A. Manukyan. [27] Further, this model was applied by Z.G. Ter-Martirosyan and A.Z. Ter-Martirosyan to solve the boundary problem with distributed loading.
Fig.2, Fig.3 are not clear and should be reproduced.
Fig.2, Fig. were reproduced.
Reviewer 2 Report
Based on previous researches, this paper provides an analytical solution for the problem of calculating nonlinear settlement and stress distribution of an excavation slopes. Then, the theoretical results are followed by numerical experiments to prove their validity. Several comments are listed as below:
1 The meaning of variables in the formula is not explained.
2 The define of materials and constitutive model used in PLAXIS2D is absent.
3 The results were not compared with that in numerical experiments.
Author Response
Based on previous researches, this paper provides an analytical solution for the problem of calculating nonlinear settlement and stress distribution of an excavation slopes. Then, the theoretical results are followed by numerical experiments to prove their validity. Several comments are listed as below:
1 The meaning of variables in the formula is not explained.
Thank you for your comments. The meaning of variables in the all formulas were added
2 The define of materials and constitutive model used in PLAXIS2D is absent.
We added information about numerical calculation in PS PLAXIS 2D. There was used the model Linear Elastic in PS PLAXIS to define isopoles of vertical and mean stresses.
3 The results were not compared with that in numerical experiments.
The calculation in the PS PLAXIS was carried out to verify isolines of vertical and mean stresses obtained earlier by analytical calculation using formulas 1-2. Long-term statement and bearing capacity were calculated using formula 13. There was no purpose to compare results in MathCAD and PLAXIS 2D. Rheological model in PLAXIS 2D is Soft Soil Creep which implements only volumetric creep deformations. We represent long-term settlement assuming that ε·z(t)= ε·v(t) + ε·γ(t) according to the Hencky’s system of physical equations using the author’s rheological model [1,27]to describe shear deformations, and another improved model, developed by Kelvin-Voigt [2] to describe volumetric deformations.
Reviewer 3 Report
The present paper states and provides an analytical solution for the problem of evaluating the settlement and load-bearing capacity of weighty soil layers of limited thickness resting upon incompressible soil bases and an excavation pit wall upon exposure of the foundation to a distributed load in the vicinity of a wall. The authors attempted to improve a mathematical model, originally established by A.Z. Ter-Martirosyan (2016) to explain shear deformations, and another improved model, developed by Kelvin-Voigt (1978) to describe volumetric deformations I went through the manuscript, I found that the conclusion has presented important information that can be a useful addition for the geotechnical practice. I suggested using more recent laboratory data to verify the proposed model. I couldn’t review the used references(data source). In general, I am satisfied with the results obtained.
Author Response
Thank you very much for your review. Additions and changes have been made to the text of the article.
Reviewer 4 Report
I have included my comments as notes on the main manuscript. Please find my attached reviewed file.

Author Response
the introduction should be improved and mention
-literature on latest developed mathematical formula for both settlement and bearing capacity for the adjacent structures for different soils
-literature on problems, techniques for excavation pits
Thank you for your comments. Authors are convinced that introduction sufficiently reveals the latest developed mathematical formula for both settlement and bearing capacity for the adjacent structures for different soils. We mentioned it giving the references [1,6,26,27]. Also it was mentioned that problems, techniques for excavation pits were described by foreign researchers [12,13,14,15,16].
for all realtions
-variables should be defined
-the sequences should be defined for the formulas
-the BCs should be clearly defined
-how about limitations?
The meaning of variables in the all formulas were added. All of the formulas have the sequences. Boundary conditions were clarified
Figures 7, 8, and 9 also should be compared with both plaxis and MathCad
There was no comparison between settlement and long-term stability obtained by formulas and settlement obtained by Plaxis because it is impossible to compare results obtained by soil model Soft Soil Creep. Rheological model in PLAXIS 2D is Soft Soil Creep which implements only volumetric creep deformations. We represent long-term settlement assuming that ε·z(t)= ε·v(t) + ε·γ(t) according to the Hencky’s system of physical equations using the author’s rheological model [1,27]to describe shear deformations, and another improved model, developed by Kelvin-Voigt [2] to describe volumetric deformations.
how about to compare the results with other existing sudies?
There is no other existing studies we could compare our results to.
also when now available software like plaxis can measure the settlement and bearing capacity of the soil for this condition then what is the main advantage of this study? how this study is importnat? does not it replicating? what is new?
The main advantage of this study is developing a method for determining the stressed state component in the reduced engineering problem based on the Ribere–Faylon trigonometric series and for accounting for the nonlinear deformation properties of soils. Also using author’s rheological model and Kelvin–Foigt’s model allows to describe creep deformations as the sum of shear creep deformations and volumetric creep deformations, assuming that ε·z(t)= ε·v(t) + ε·γ(t) , according to the Henky’s system of physical equations. А. Z. Ter-Martirosyan proposed a new rheological equation allowing to plot the γ - t dependence at different τ as a double curvature curve. Also, purposed author’s rheological model was verified by laboratory tests, which was carried out by A. Z. Ter-Martirosyan, L. Yu. Ermoshina and A. Manukyan. [27] This dependence corresponds to clay soils tests, conducted by S.S. Vyalov, S.R. Meschyan and others in last century.
Round 2
Reviewer 2 Report
Accept in present form
Author Response
Thanks for the previous comments that helped improve the quality of the article.
Reviewer 4 Report
Please find my new comments based on your responses in the attached and uploaded file.

Author Response
NEW Comment regarding this comment
Which software can do the comparison with your developed formula (FLAC OR SO-Foundation or any other) with available soil model defined for
-settlement
-long term stability
Please explain and use that software instead of Plaxis
For long term stability of the soil, it is the function of so many parameters. Which parameter you are focusing in your proposed model that is affecting the soil?!
It was mentioned in article that the results of our calculations were obtained by PS MathCAD. We focused on rheological parameters of the soil, we didn't have a goal to take into account all the soil properties. We don’t have to use other software because author’s rheological model was verified by laboratory tests, which was carried out by A. Z. Ter-Martirosyan, L. Yu. Ermoshina and A. Manukyan. [27] And according to the results of clay soils tests, conducted by S.S. Vyalov, S.R. Meschyan and others [2,21,22], it was found that the diagram of non-reducing creep deformations was typical for soils, having rheological properties and subjected to high stresses in the case of long-term loading.
Additionally, it should be noted that in the future the authors plan to embed the described soil model in Plaxis, which will make it possible to perform comparative calculations. However, this part is a separate research task.
NEW Comment
Fig. 6 a) and b), it does not have vertical and horizontal axis titles with units. Please update.
Updated
NEW Comment
Fig. 9b), how did you come up with loading selection?
The loadings p1, p2, p3, p4, p5, p6 were selected in accordance with the loads from the 6-storey, 10-storey, 12-storey, 14-storey, 15-storey, 16-storey buildings. This types of residential buildings are widespread in the cities of our country
NEW Comment
Line 68-77. Why should we mention researchers by nationality?! Is it important in the science and research community? What is the goal of research?! If the ultimate goal is contributing to all human condition improvement, then it is not important at all, if you think it matters, what is it?!
Based on
Research and Publication Ethics
Research Ethics
Research Involving Human Subjects
Additionally, when studies describe groups by race, ethnicity, gender, disability, disease, etc., explanation regarding why such categorization was needed must be clearly stated in the article.
Please explain.
We apologize for mentioning the Russian nationality of scientists. We had no purpose to violate journalistic ethics. Changes made in the article.
NEW Comment
Also, most of the sentences are really hard to understand because you tried to tell the point with so many details in only one sentence. It is really hard to get the point. Most of the sentences should be broken into different sub sentences.
We understand that some sentences are difficult to understand, but dividing large sentences into several simple ones will increase the volume of the article. We tried to briefly outline the essence of a large and complex research
Round 3
Reviewer 4 Report
Tha Authors addressed the comments properly.